# Replacing Rewards with Examples: Example-Based Policy Search via Recursive Classification

**Benjamin Eysenbach**[1,2]   **Sergey Levine**[2,3]   **Ruslan Salakhutdinov**[1]
[1]Carnegie Mellon University,   [2]Google Brain,   [3]UC Berkeley
beysenba@cs.cmu.edu

## Abstract

Reinforcement learning (RL) algorithms assume that users specify tasks by manually writing down a reward function. However, this process can be laborious and demands considerable technical expertise. Can we devise RL algorithms that instead enable users to specify tasks simply by providing examples of successful outcomes? In this paper, we derive a control algorithm that maximizes the future probability of these successful outcome examples. Prior work has approached similar problems with a two-stage process, first learning a reward function and then optimizing this reward function using another RL algorithm. In contrast, our method directly learns a value function from transitions and successful outcomes, without learning this intermediate reward function. Our method therefore requires fewer hyperparameters to tune and lines of code to debug. We show that our method satisfies a new data-driven Bellman equation, where examples take the place of the typical reward function term. Experiments show that our approach outperforms prior methods that learn explicit reward functions.[1]

## 1   Introduction

In supervised learning settings, tasks are defined by data: what causes a car detector to detect cars is not the choice of loss function (which might be the same as for an airplane detector), but the choice of training data. Defining tasks in terms of data, rather than specialized loss functions, arguably makes it easier to apply machine learning algorithms to new domains. In contrast, reinforcement learning (RL) problems are typically posed in terms of reward functions, which are typically manually designed. Arguably, the challenge of designing reward functions has limited RL to applications with simple reward functions, and has been restricted to users who speak this language of mathematically-defined reward functions. Can we make task specification in RL similarly data-driven?

Whereas the standard MDP formalism centers around predicting and maximizing the future reward, we will instead focus on the problem classifying whether a task will be solved in the future. The user will provide a collection of example success states, not a reward function. We call this problem setting *example-based control*. In effect, these examples tell the agent "What would the world look like if the task were solved?" For example, for the task of opening a door, success examples correspond to different observations of the world when the door is open. The user can find examples of success even for tasks that they themselves do not know how to solve. For example, the user could solve the task using actions unavailable to the agent (e.g., the user may have two arms, but a robotic agent may have only one) or the user could find success examples by searching the internet. As we will discuss in Sec. 3.1, this problem setting is different from imitation learning: we maximize a different objective function and only assume access to success examples, not entire expert trajectories.

---

[1]Project site with videos and code: `https://ben-eysenbach.github.io/rce`

35th Conference on Neural Information Processing Systems (NeurIPS 2021).

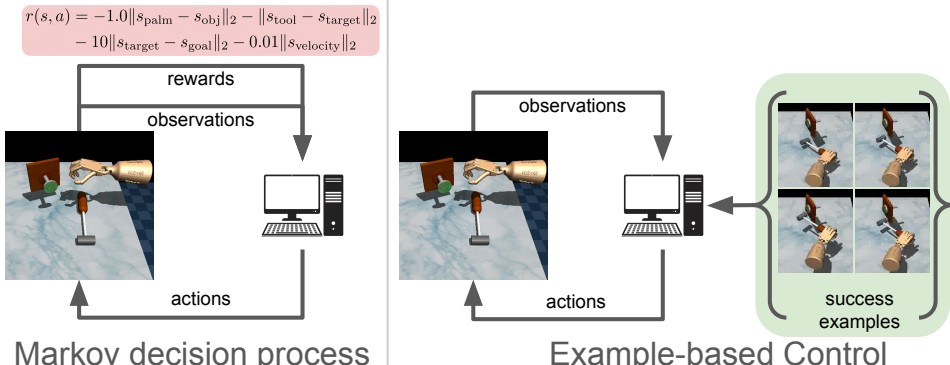

$$r(s,a) = -1.0\|s_{\text{palm}} - s_{\text{obj}}\|_2 - \|s_{\text{tool}} - s_{\text{target}}\|_2$$
$$- 10\|s_{\text{target}} - s_{\text{goal}}\|_2 - 0.01\|s_{\text{velocity}}\|_2$$

Figure 1: **Example-based control**: Whereas the standard MDP framework requires a user-defined reward function, example-based control specifies tasks via a handful of user-provided success examples.

Learning from examples is challenging because we must automatically identify when the agent has solved the task and reward it for doing so. Prior methods (either imitation learning from demonstrations or learning from success examples) take an indirect approach that resembles inverse RL: first learn a separate model to represent the reward function, and then optimize this reward function with standard RL algorithms. Our method is different from these prior methods because it learns to predict *future* success directly from transitions and success examples, without learning a separate reward function. This key difference has important algorithmic, theoretical, and empirical benefits. **Algorithmically**, our end-to-end approach removes potential biases in learning a separate reward function, reduces the number of hyperparameters, and simplifies the resulting implementation. **Theoretically**, we propose a method for classifying *future* events using a variant of temporal difference learning that we call recursive classification. This method satisfies a new Bellman equation, where success examples are used in place of the standard reward function term. We use this result to provide convergence guarantees. **Empirically**, we demonstrate that our method solves many complex manipulation tasks that prior methods fail to solve.

Our paper also addresses a subtle but important ambiguity in formulating example-based control. Some states might always solve the task while other states might rarely solve the task. But, without knowing how often the user visited each state, we cannot determine the likelihood that each state solves the task. Thus, an agent can only estimate the probability of success by making an additional assumption about how the success examples were generated. We will discuss two choices of assumptions. The first choice of assumption is convenient from an algorithmic perspective, but is sometimes violated in practice. A second choice is a worst-case approach, resulting in a problem setting that we call *robust example-based control*. Our analysis shows that the robust example-based control objective is equivalent to minimizing the squared Hellinger distance (an $f$-divergence).

In summary, this paper studies a data-driven framing of control, where reward functions are replaced by examples of successful outcomes. Our main contribution is an algorithm for off-policy example-based control. The key idea of the algorithm is to directly learn to predict whether the task will be solved in the future via *recursive classification*, without using separate reward learning and policy search procedures. Our analysis shows that our method satisfies a new Bellman equation where rewards are replaced by data (examples of success). Empirically, our method significantly outperforms state-of-the-art imitation learning methods (AIRL [7], DAC [17], and SQIL [28]) and recent methods that learn reward functions (ORIL [41], PURL [38], and VICE [8]). Our method completes complex tasks, such as picking up a hammer to knock a nail into a board, tasks that *none* of these baselines can solve. Using tasks with image observations, we demonstrate agents learned with our method acquire a notion of success that generalizes to new environments with varying shapes and goal locations.

## 2  Related Work

**Learning reward functions.**  Prior works have studied RL in settings where the task is specified either with examples of successful outcomes or complete demonstrations. These prior methods typically learn a reward function from data and then apply RL to this reward function (e.g., Fu et al. [8], Ziebart et al. [40]). Most inverse RL algorithms adopt this approach [1, 25, 27, 29, 36, 40], as do more recent methods that learn a *success classifier* to distinguishing successful outcomes from random states [8, 16, 31, 41]. Prior adversarial imitation learning methods [7, 12] can be

viewed as *iteratively* learning a success classifier. Recent work in this area focuses on extending these methods to the offline setting [16, 41], incorporating additional sources of supervision [42], and learning the classifier via positive-unlabeled classification [13, 38, 41]. Many prior methods for robot learning have likewise used a classifier to distinguish success examples [2, 22, 34, 37]. Unlike these prior methods, our approach only requires examples of successful outcomes (not expert trajectories) and does not learn a separate reward function. Instead, our method learns a value function directly from examples, effectively "cutting out the middleman." This difference from prior work removes hyperparameters and potential bugs associated with learning a success classifier. Empirically, we demonstrate that our end-to-end approach outperforms these prior two-stage approaches. See Appendix D for more discussion of the relationship between our method and prior work.

**Imitation learning without auxiliary classifiers.**   While example-based control is different from imitation learning, our method is similar to two prior imitation learning methods that likewise avoid learning a separate reward function [18, 28]. ValueDICE [18], a method based on convex duality, uses full expert demonstrations for imitation learning. In contrast, our method learns from success examples, which are typically easier to provide than full expert demonstrations. SQIL [28] is a modification of SAC [11] that labels success examples with a reward of $+1$. The mechanics of our method are similar to SQIL [28], but key algorithmic differences (backed by stronger theoretical guarantees) result in better empirical performance. Our analysis in Sec. 4.2 highlights connections and differences between imitation learning and example-based control.

**Goal-conditioned RL.**   Goal-conditioned RL provides one way to specify tasks in terms of data, and prior work has shown how goal-conditioned policies can be learned directly from data, without a reward function [4, 15, 21, 30]. However, the problem that we study in this paper, example-based control, is different from goal-conditioned RL because it allows users to indicate that tasks can be solved in many ways, enabling the agent to learn a more general notion of success. Perhaps the most similar prior work in this area is C-learning [4], which uses a temporal-difference update to learn the probability that a goal state will be reached in the future. Our method will use a similar temporal-difference update to predict whether *any* success example will be reached in the future. Despite the high-level similarity with C-learning, our algorithm will be markedly different; for example, our method does not require hindsight relabeling, and learns a single policy rather than a goal-conditioned policy.

## 3   Example-Based Control via Recursive Classification

We aim to learn a policy that reaches states that are likely to solve the task (see Fig. 1), without relying on a reward function. We start by formally describing this problem, which we will call *example-based control*. We then propose a method for solving this problem and provide convergence guarantees.

### 3.1   Problem Statement

Example-based control is defined by a controlled Markov process (i.e., an MDP without a reward function) with dynamics $p(\mathbf{s_{t+1}} \mid \mathbf{s_t}, \mathbf{a_t})$ and an initial state distribution $p_1(\mathbf{s_1})$, where $\mathbf{s_t} \in \mathcal{S}$ and $\mathbf{a_t}$ denote the time-indexed states and actions. The variable $\mathbf{s_{t+\Delta}}$ denotes a state $\Delta$ steps in the future.

The agent is given a set of *success examples*, $\mathcal{S}^* = \{\mathbf{s}^*\} \subseteq \mathcal{S}$. The random variable $\mathbf{e_t} \in \{0, 1\}$ indicates whether the task is solved at time $t$, and $p(\mathbf{e_t} \mid \mathbf{s_t})$ denotes the probability that the current state $\mathbf{s_t}$ solves the task. Given a policy $\pi_\phi(\mathbf{a_t} \mid \mathbf{s_t})$, we define the discounted future state distribution:

$$p^\pi(\mathbf{s_{t+}} \mid \mathbf{s_t}, \mathbf{a_t}) \triangleq (1 - \gamma) \sum_{\Delta=0}^{\infty} \gamma^\Delta p^\pi(\mathbf{s_{t+\Delta}} = \mathbf{s_{t+}} \mid \mathbf{s_t}, \mathbf{a_t}). \tag{1}$$

Using this definition, we can write the probability of solving the task at a *future* step as

$$p^\pi(\mathbf{e_{t+}} \mid \mathbf{s_t}, \mathbf{a_t}) \triangleq \mathbb{E}_{p^\pi(\mathbf{s_{t+}}\mid\mathbf{s_t},\mathbf{a_t})}[p(\mathbf{e_{t+}} \mid \mathbf{s_{t+}})]. \tag{2}$$

Example-based control maximizes the probability of solving the task in the (discounted) future:

**Definition 1** (Example-based control). *Given a controlled Markov process and distribution over success examples $p(\mathbf{s_t} \mid \mathbf{e_t} = 1)$, the example-based control problem is to find the policy that optimizes the likelihood of solving the task:*

$$\arg\max_\pi p^\pi(\mathbf{e_{t+}} = 1) = \mathbb{E}_{p_1(\mathbf{s_1}),\pi(\mathbf{a_1}\mid\mathbf{s_1})} [p^\pi(\mathbf{e_{t+}} = 1 \mid \mathbf{s_1}, \mathbf{a_1})].$$

Although this objective is equivalent to the RL objective with rewards $r(\mathbf{s_t}, \mathbf{a_t}) = p(\mathbf{e_t} = 1 \mid \mathbf{s_t})$, we assume the probabilities $p(\mathbf{e_t} \mid \mathbf{s_t})$ are unknown. Instead, we assume that we have samples of successful states, $\mathbf{s}^* \sim p_U(\mathbf{s_t} \mid \mathbf{e_t} = 1)$. Example-based control differs from imitation learning because imitation learning requires full expert demonstrations. In the special case where the user provides a single success state, example-based control is equivalent to goal-conditioned RL.

Since interacting with the environment to collect experience is expensive in many settings, we define *off-policy example-based control* as the version of this problem where the agent learns from environment interactions collected from other policies. In this setting, the agent learns from two distinct datasets: **(1)** transitions, $\{(\mathbf{s_t}, \mathbf{a_t}, \mathbf{s_{t+1}}) \sim p_U(\mathbf{s_t}, \mathbf{a_t}, \mathbf{s_{t+1}})\}$, which contain information about the environment dynamics; and **(2)** success examples, $\mathcal{S}^* = \{\mathbf{s}^* \sim p_U(\mathbf{s_t} \mid \mathbf{e_t} = 1)\}$, which specify the task that the agent should attempt to solve. Our analysis will assume that these two datasets are fixed. The main contribution of this paper is an algorithm for off-policy example-based control.

**An assumption on success examples.** The probability of solving the task at state $\mathbf{s_t}$, $p(\mathbf{e_t} = 1 \mid \mathbf{s_t})$, cannot be uniquely determined from success examples and transitions alone. To explain this ambiguity, we define $p_U(\mathbf{s_t})$ as the state distribution visited by the user; note that the user may be quite bad at solving the task themselves. Then, the probability of solving the task at state $\mathbf{s_t}$ depends on how often a user visits state $\mathbf{s_t}$ versus how often the task is solved when visiting state $\mathbf{s_t}$:

$$p(\mathbf{e_t} = 1 \mid \mathbf{s_t}) = \frac{p_U(\mathbf{s_t} \mid \mathbf{e_t} = 1)}{p_U(\mathbf{s_t})} p_U(\mathbf{e_t} = 1). \tag{3}$$

For example, the user may complete a task using two strategies, but we cannot determine which of these strategies is more likely to succeed unless we know how often the user attempted each strategy. Thus, any method that learns from success examples *must* make an additional assumption on $p_U(\mathbf{s_t})$. We will discuss two choices of assumptions. The **first choice** is to assume that the user visited states with the same frequency that they occur in the dataset of transitions. That is,

$$p_U(\mathbf{s_t}) = \iint p_U(\mathbf{s_t}, \mathbf{a_t}, \mathbf{s_{t+1}}) d\mathbf{a_t} d\mathbf{s_{t+1}}. \tag{4}$$

Intuitively, this assumption implies that the user has the same capabilities as the agent. Prior work makes this same assumption without stating it explicitly [8, 24, 31]. Experimentally, we find that our method succeeds even in cases where this assumption is violated.

However, many common settings violate this assumption, especially when the user has different dynamics constraints than the agent. For example, a human user collecting success examples for a cleaning task might usually put away objects on a shelf at eye-level, whereas transitions collected by a robot interact with the ground-level shelves more frequently. Under our previous assumption, the robot would assume that putting objects away on higher shelves is more satisfactory than putting them away on lower shelves, even though doing so might be much more challenging for the robot. To handle these difference in capabilities, the **second choice** is to use a *worst-case* formulation, which optimizes the policy to be robust to *any* choice of $p_U(\mathbf{s_t})$. Surprisingly, this setting admits a tractable solution, as we discuss in Sec. 4.2.

### 3.2 Predicting Future Success by Recursive Classification

We now describe our method for example-based control. We start with the more standard **first choice** for the assumption on $p_U(\mathbf{s_t})$ (Eq. 4); we discuss the second choice in Sec. 4.2. Our approach estimates the probability in Eq. 2 indirectly via a future success classifier. This classifier, $C_\theta^\pi(\mathbf{s_t}, \mathbf{a_t})$, discriminates between "positive" state-action pairs which lead to successful outcomes (i.e., sampled from $p(\mathbf{s_t}, \mathbf{a_t} \mid \mathbf{e_{t+}} = 1)$) and random "negatives" (i.e., sampled from the marginal distribution $p(\mathbf{s_t}, \mathbf{a_t})$). We will use different class-specific weights, using a weight of $p(\mathbf{e_{t+}} = 1)$ for the "positives" and a weight of 1 for the "negatives." Bayes-optimal classifier is

$$C_\theta^\pi(\mathbf{s_t}, \mathbf{a_t}) = \frac{p^\pi(\mathbf{s_t}, \mathbf{a_t} \mid \mathbf{e_{t+}} = 1)p(\mathbf{e_{t+}} = 1)}{p^\pi(\mathbf{s_t}, \mathbf{a_t} \mid \mathbf{e_{t+}} = 1)p(\mathbf{e_{t+}} = 1) + p(\mathbf{s_t}, \mathbf{a_t})}. \tag{5}$$

These class specific weights let us predict the probability of future success using the optimal classifier:

$$\frac{C_\theta^\pi(\mathbf{s_t}, \mathbf{a_t})}{1 - C_\theta^\pi(\mathbf{s_t}, \mathbf{a_t})} = p^\pi(\mathbf{e_{t+}} = 1 \mid \mathbf{s_t}, \mathbf{a_t}). \tag{6}$$

---

**Algorithm 1** Recursive Classification of Examples

---

**Input**: success examples $\mathcal{S}^*$
Initialize policy $\pi_\phi(\mathbf{a_t} \mid \mathbf{s_t})$, classifier $C_\theta^\pi(\mathbf{s_t}, \mathbf{a_t})$, replay buffer $\mathcal{D}$
**while** not converged **do**
    Collect a new trajectory: $\mathcal{D} \leftarrow \mathcal{D} \cup \{\tau \sim \pi_\phi\}$
    Sample success examples: $\{\mathbf{s_t}^{(1)} \sim \mathcal{S}^*, \mathbf{a_t}^{(1)} \sim \pi_\phi(\mathbf{a_t} \mid \mathbf{s_t}^{(1)})\}$
    Sample transitions: $\{(\mathbf{s_t}^{(2)}, \mathbf{a_t}^{(2)}, \mathbf{s_{t+1}}) \sim \mathcal{D}, \mathbf{a_{t+1}} \sim \pi_\phi(\mathbf{a_{t+1}} \mid \mathbf{s_{t+1}})\}$
    $w \leftarrow \frac{C_\theta^\pi(\mathbf{s_{t+1}}, \mathbf{a_{t+1}})}{1 - C_\theta^\pi(\mathbf{s_{t+1}}, \mathbf{a_{t+1}})}$                                                               ▷ Eq. 9
    $\mathcal{L}(\theta) \leftarrow (1-\gamma)\mathcal{CE}(C_\theta(\mathbf{s_t}^{(1)}, \mathbf{a_t}^{(1)}); y=1) + (1+\gamma w)\mathcal{CE}(C_\theta(\mathbf{s_t}^{(2)}, \mathbf{a_t}^{(2)}); y=\frac{\gamma w}{1+\gamma w})$
    Update classifier: $\theta \leftarrow \theta + \eta \nabla_\theta \mathcal{L}(\theta)$                                                  ▷ Eq. 8
    Update policy: $\phi \leftarrow \phi + \eta \nabla_\phi \mathbb{E}_{\pi_\phi(\mathbf{a_t}|\mathbf{s_t})}[C_\theta(\mathbf{s_t}, \mathbf{a_t})]$
**return** $\pi_\phi$

---

Importantly, the resulting method will not actually require estimating the weight $p(\mathbf{e_{t+}} = 1)$. We would like to optimize the classifier parameters using maximum likelihood estimation:

$$\mathcal{L}^\pi(\theta) \triangleq p(\mathbf{e_{t+}} = 1) \, \mathbb{E}_{p(\mathbf{s_t}, \mathbf{a_t}|\mathbf{e_{t+}}=1)}[\log C_\theta^\pi(\mathbf{s_t}, \mathbf{a_t})] + \mathbb{E}_{p(\mathbf{s_t}, \mathbf{a_t})}[\log(1 - C_\theta^\pi(\mathbf{s_t}, \mathbf{a_t}))]. \quad (7)$$

However, we cannot directly optimize this objective because we cannot sample from $p(\mathbf{s_t}, \mathbf{a_t} \mid \mathbf{e_{t+}} = 1)$. We convert Eq. 7 into an equivalent loss function that we can optimize using three steps; see Appendix A for a detailed derivation. The **first step** is to factor the distribution $p(\mathbf{s_t}, \mathbf{a_t}, \mathbf{e_{t+}} = 1)$. The **second step** is to decompose $p^\pi(\mathbf{e_{t+}} = 1 \mid \mathbf{s_t}, \mathbf{a_t})$ into two terms, corresponding to the probabilities of solving the task at time $t' = t + 1$ and time $t' > t + 1$. We can estimate the probability of solving the task at the next time step using the set of success examples. The **third step** is to estimate the probability of solving the task at time $t' > t + 1$ by evaluating the classifier at the next time step. Combining these three steps, we can *equivalently* express the objective function in Eq. 7 using off-policy data:

$$\mathcal{L}^\pi(\theta) = (1-\gamma)\mathbb{E}_{\substack{p_U(\mathbf{s_t}|\mathbf{e_t}=1) \\ p(\mathbf{a_t}|\mathbf{s_t})}}[\underbrace{\log C_\theta^\pi(\mathbf{s_t}, \mathbf{a_t})}_{(a)}] + \mathbb{E}_{p(\mathbf{s_t}, \mathbf{a_t}, \mathbf{s_{t+1}})}[\underbrace{\gamma w \log C_\theta^\pi(\mathbf{s_t}, \mathbf{a_t})}_{(b)} + \underbrace{\log(1 - C_\theta^\pi(\mathbf{s_t}, \mathbf{a_t}))}_{(c)}],$$

$$(8)$$

where

$$w = \mathbb{E}_{p(\mathbf{a_{t+1}}|\mathbf{s_{t+1}})}\left[\frac{C_\theta^\pi(\mathbf{s_{t+1}}, \mathbf{a_{t+1}})}{1 - C_\theta^\pi(\mathbf{s_{t+1}}, \mathbf{a_{t+1}})}\right] \quad (9)$$

is the classifier's prediction (ratio) at the next time step. Our resulting method can be viewed as a temporal difference [33] approach to classifying future events. We will refer to our method as **recursive classification of examples (RCE)**. This equation has an intuitive interpretation. The first term *(a)* trains the classifier to predict 1 for the success examples themselves, and the third term *(c)* trains the classifier to predict 0 for random transitions. The important term is the second term *(b)*, which is analogous to the "bootstrapping" term in temporal difference learning [32]. Term *(b)* indicates that the probability of future success depends on the probability of success at the *next* time step, as inferred using the classifier's own predictions.

Our resulting method is similar to existing actor-critic RL algorithms. To highlight the similarity to existing actor-critic methods, we can combine the *(b)* and *(c)* terms in the classifier objective function (Eq. 8) to express the loss function in terms of two cross entropy losses:

$$\min_\theta (1-\gamma)\mathbb{E}_{\substack{p(\mathbf{s_t}|\mathbf{e_t}=1), \\ \mathbf{a_t} \sim \pi(\mathbf{a_t}|\mathbf{s_t})}}[\mathcal{CE}(C_\theta^\pi(\mathbf{s_t}, \mathbf{a_t}); y=1)] + (1+\gamma w)\mathbb{E}_{p(\mathbf{s_t}, \mathbf{a_t}, \mathbf{s_{t+1}})}\left[\mathcal{CE}\left(C_\theta^\pi(\mathbf{s_t}, \mathbf{a_t}); y=\frac{\gamma w}{\gamma w + 1}\right)\right]. \quad (10)$$

These cross entropy losses update the classifier to predict $y = 1$ for the success examples and to predict $y = \frac{\gamma w}{1+\gamma w}$ for other states.

**Algorithm summary.** Alg. 1 summarizes our method, which alternates between updating the classifier, updating the policy, and (optionally) collecting new experience. We update the policy to choose actions that maximize the classifier's confidence that the task will be solved in the future: $\max_\phi \mathbb{E}_{\pi_\phi(\mathbf{a_t}|\mathbf{s_t})}[C_\theta^\pi(\mathbf{s_t}, \mathbf{a_t})]$. Following prior work [5, 35]), we regularized the policy updates by adding an entropy term with coefficient $\alpha = 10^{-4}$. We also found that using N-step returns significantly improved the results of RCE (see Appendix F for details and ablation experiments.). Implementing our method on top of existing methods such as SAC [11] or TD3 [9] requires only changing the standard Bellman loss with the loss in Eq. 10. See Appendix E for implementation details; code is available on the project website.

# 4 Analysis

In this section, we prove that RCE satisfies many of the same convergence and optimality guarantees (for example-based control) that standard RL algorithms satisfy (for reward-based MDPs). These results are important as they demonstrate that formulating control in terms of data, rather than rewards, does preclude algorithms from enjoying strong theoretical guarantees. Proofs of all results are given in Appendix B, and we include a further discussion of how RCE relates to prior work in Appendix D.

## 4.1 Bellman Equations and Convergence Guarantees

To prove that RCE converges to the optimal policy, we will first show that RCE satisfies a new Bellman equation:

**Lemma 4.1.** *The Bayes-optimal classifier $C^\pi$ for policy $\pi$ satisfies the following identity:*

$$\frac{C^\pi(\mathbf{s_t}, \mathbf{a_t})}{1 - C^\pi(\mathbf{s_t}, \mathbf{a_t})} = (1-\gamma)p(\mathbf{e_t} = 1 \mid \mathbf{s_t}) + \gamma \mathbb{E}_{\substack{p(\mathbf{s_{t+1}}|\mathbf{s_t}, \mathbf{a_t}) \\ \pi(\mathbf{a_{t+1}}|\mathbf{s_{t+1}})}} \left[ \frac{C^\pi(\mathbf{s_{t+1}}, \mathbf{a_{t+1}})}{1 - C^\pi(\mathbf{s_{t+1}}, \mathbf{a_{t+1}})} \right]. \quad (11)$$

The proof combines the definition of the Bayes-optimal classifier with the assumption from Eq. 4. This Bellman equation is analogous to the standard Bellman equation for Q-learning, where the reward function is replaced by $(1 - \gamma)p(\mathbf{e_t} = 1 \mid \mathbf{s_t})$ and the Q function is parametrized as $Q_\theta^\pi(\mathbf{s_t}, \mathbf{a_t}) = \frac{C_\theta^\pi(\mathbf{s_t}, \mathbf{a_t})}{1 - C_\theta^\pi(\mathbf{s_t}, \mathbf{a_t})}$. While we do not know how to compute this reward function, the update rule for RCE is equivalent to doing value iteration using that reward function and that parametrization of the Q-function:

**Lemma 4.2.** *In the tabular setting, the **expected** updates for RCE are equivalent to doing value iteration with the reward function $r(\mathbf{s_t}) = (1 - \gamma)p(\mathbf{e_t} = 1 \mid \mathbf{s_t})$ and a Q-function parametrized as $Q_\theta^\pi(\mathbf{s_t}, \mathbf{a_t}) = \frac{C_\theta^\pi(\mathbf{s_t}, \mathbf{a_t})}{1 - C_\theta^\pi(\mathbf{s_t}, \mathbf{a_t})}$.*

This result tells us that RCE is equivalent to maximizing the reward function $(1 - \gamma)p(\mathbf{e_t} = 1 \mid \mathbf{s_t})$; however, RCE does not require knowing $p(\mathbf{e_t} = 1 \mid \mathbf{s_t})$, the probability that each state solves the task. Since value iteration converges in the tabular setting, an immediate consequence of Lemma 4.2 is that tabular RCE also converges:

**Corollary 4.2.1.** *RCE converges in the tabular setting.*

So far we have analyzed the training process for the classifier for a fixed policy. We conclude this section by showing that optimizing the policy w.r.t. the classifier improves the policy's performance.

**Lemma 4.3.** *Let policy $\pi(\mathbf{a_t} \mid \mathbf{s_t})$ and success examples $\mathcal{S}^*$ be given, and let $C^\pi(\mathbf{s_t}, \mathbf{a_t})$ denote the corresponding Bayes-optimal classifier. Define the improved policy as acting greedily w.r.t. $C^\pi$: $\pi'(\mathbf{a_t} \mid \mathbf{s_t}) = \mathbb{1}(\mathbf{a} = \arg\max_a C^\pi(\mathbf{s_t}, \mathbf{a}))$. Then the improved policy is at least as good as the old policy at solving the task: $p^{\pi'}(\mathbf{e_{t+}} = 1) \geq p^\pi(\mathbf{e_{t+}} = 1)$.*

## 4.2 Robust Example-based Control

In this section, we derive a principled solution for the case where $p_U(\mathbf{s_t})$ is not known, which will correspond to modifying the objective function for example-based control. However, we will argue that, in some conditions, the method proposed in Sec. 3.2 is *already* robust to unknown $p_U(\mathbf{s_t})$, if that method is used with online data collection. The goal of this discussion is to provide a theoretical relationship between our method and a robust version of example-based control that makes fewer assumptions about $p_U(\mathbf{s_t})$. This discussion will also clarify how changing assumptions on the user's capabilities can change the optimal policy.

When introducing example-based control in Sec. 3.1, we emphasized that we *must* make an assumption to make the example-based control problem well defined. The exact probability that a success example solves the task depends on how often the user visited that state, which the agent does not know. Therefore, there are many valid hypotheses for how likely each state is to solve the task. We can express the set of valid hypotheses using Bayes' Rule:

$$\mathcal{P}_{\mathbf{e_t}|\mathbf{s_t}} \triangleq \left\{ \hat{p}(\mathbf{e_t} = 1 \mid \mathbf{s_t}) = \frac{p_U(\mathbf{s_t} \mid \mathbf{e_t} = 1)p(\mathbf{e_t} = 1)}{p_U(\mathbf{s_t})} \right\}.$$

Previously (Sec. 3.2), we resolved this ambiguity by assuming that $p_U(\mathbf{s_t})$ was equal to the distribution over states in our dataset of transitions. As discussed in Sec. 3.1, many problem settings violate this assumption, prompting us to consider the more stringent setting with no prior information about $p_U(\mathbf{s_t})$ (e.g., no prior knowledge about the user's capabilities). To address this setting, we will assume the *worst* possible choice of $p_U(\mathbf{s_t})$. This approach will make the agent robust to imperfect knowledge of the user's abilities and to mislabeled success examples. Formally, we define the *robust example-based control* problem as

$$\max_{\pi} \min_{\hat{p}(\mathbf{e_t}|\mathbf{s_t}) \in \mathcal{P}_{\mathbf{e_t}|\mathbf{s_t}}} \mathbb{E}_{p^{\pi}(\mathbf{s_{t+}})}[\hat{p}(\mathbf{e_{t+}} = 1 \mid \mathbf{s_{t+}})] = \max_{\pi} \min_{p_U(\mathbf{s_t})} \mathbb{E}_{p^{\pi}(\mathbf{s_{t+}})}\left[\frac{p_U(\mathbf{s_t} \mid \mathbf{e_t} = 1)}{p_U(\mathbf{s_t})}p(\mathbf{e_t} = 1)\right].$$
(12)

This objective can be understood as having the adversary assign a weight of $1/p_U(\mathbf{s_t})$ to each success example. The optimal adversary will assign lower weights to success examples that the policy frequently visits and higher weights to less-visited success examples. Intuitively, the optimal policy should try to reach many of the success examples, not just the ones that are easiest to reach. Thus, such a policy will continue to succeed even if certain success examples are removed, or are later discovered to have been mislabeled. Surprisingly, solving this two-player game corresponds to minimizing an $f$-divergence:

**Lemma 4.4.** *Define $H^2[p(\mathbf{x}), q(\mathbf{x})] = \int(\sqrt{p(\mathbf{x})} - \sqrt{q(\mathbf{x})})^2 d\mathbf{x}$ as the squared Hellinger distance, an $f$-divergence. Robust example-based control (Eq. 12) is equivalent to minimizing the squared Hellinger distance between policy's discounted state occupancy measure and the **conditional** distribution $p(\mathbf{s_t} \mid \mathbf{e_t} = 1)$:*

$$\min_{\hat{p}(\mathbf{e_t}|\mathbf{s_t}) \in \mathcal{P}_{\mathbf{e_t}|\mathbf{s_t}}} p^{\pi,\hat{p}}(\mathbf{e_{t+}}) = 1 - \frac{1}{2}H^2[p(\mathbf{s_t}|\mathbf{e_t} = 1), p^{\pi}(\mathbf{s_{t+}} = \mathbf{s_t})].$$

The main idea of the proof (found in Appendix C) is to compute the worst-case distribution $p_U(\mathbf{s_t})$ using the calculus of variations. Preliminary experiments (Fig. 5 in Appendix C) show that a version of RCE with online data collection finds policies that perform well on the robust example-based control objective (Eq. 12). In fact, under somewhat stronger assumptions, we can show that the solution of robust example-based control is a fixed point of *iterated* RCE (see Appendix C.2). Therefore, in our experiments, we use RCE with online data collection.

## 5 Experiments

Our experiments study how effectively RCE solves example-based control tasks, especially in comparison to prior methods that learn an explicit reward function. Both RCE and the prior methods receive only the success examples as supervision; no method has access to expert trajectories of reward functions. Additional experiments in Sec. 5.2 study whether RCE can solve tasks using image observations. These experiments test whether RCE can solve tasks in new environments that are different from those where the success examples were collected, and test whether RCE learns policies that learn a general notion of success rather than just memorizing the success examples. We include videos of learned policies online[2] and include implementation details, hyperparameters, ablation experiments, and a list of failed experiments in the Appendix.

We compare RCE against prior methods that infer a reward function from the success examples and then apply an off-the-shelf RL algorithm; some baselines iterate between these two steps. AIRL [7] is a popular adversarial imitation learning method. VICE [8] is the same algorithm as AIRL, but intended to be applied to success examples rather than full demonstrations. We will label this method as "VICE" in figures, noting that it is the same algorithm as AIRL. DAC [17] is a more recent, off-policy variant of AIRL. We also compared against two recent methods that learn rewards from *demonstrations*: ORIL [41] and PURL [38]. Following prior work [16], we also compare against "frozen" variants of some baselines that first train the parameters of the reward function and then apply RL to that reward function without updating the parameters of the reward function again. Our method differs from these baselines in that we do not learn a reward function from the success examples and then apply RL, but rather learn a policy directly from the success examples. Lastly, we compare against SQIL [28], an imitation learning method that assigns a reward of $+1$ to states from demonstrations and $0$ to all other states. SQIL does not learn a separate reward function and structurally resembles our method, but is derived from different principles (see Sec. 2.).

---

[2]`https://ben-eysenbach.github.io/rce`

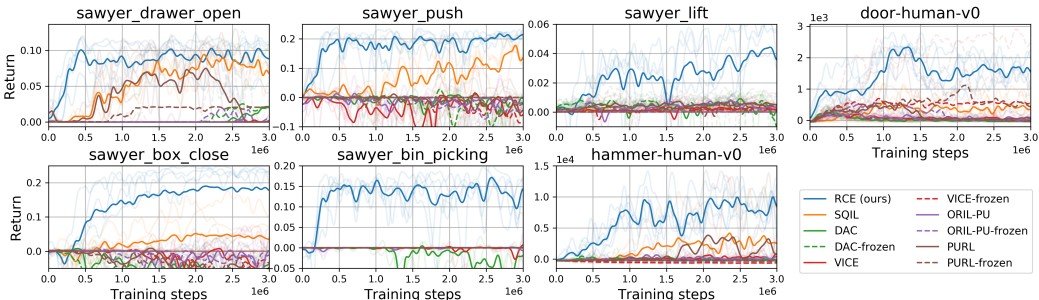

Figure 2: **Recursive Classification of Examples for learning manipulation tasks**: We apply RCE to a range of manipulation tasks, each accompanied with a dataset of success examples. For example, on the `sawyer_lift` task, we provide success examples where the object has been lifted above the table. We use the cumulative task return (↑ is better) solely for evaluation. Our method (blue line) outperforms prior methods across all tasks.

## 5.1 Evaluating RCE for Example-Based Control.

We evaluate each method on five Sawyer manipulation tasks from Meta-World [39] and two manipulation tasks from Rajeswaran et al. [26]. Fig. 3 illustrates these tasks. On each task, we provide the agent with 200 successful outcomes to define the task. For example, on the `open_drawer` task, these success examples show an opened drawer. As another example, on the `sawyer_push` task, success examples not only have the end effector touching the puck, but (more importantly) the puck position is different.

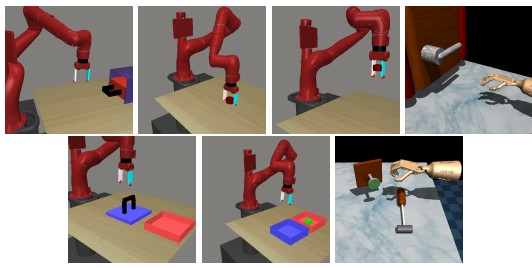

Figure 3: **Manipulation Environments**

We emphasize that these success examples only reflect the final state where the task is solved and are not full expert trajectories. This setting is important in practical use-cases: it is often easier for humans to arrange the workspace into a successful configuration than it is to collect an entire demonstration. See Appendix E.3 for details on how success examples were generated for each task. While these tasks come with existing user-defined reward functions, these rewards are not provided to any of the methods in our experiments and are used solely for evaluation (↑ is better). We emphasize that this problem setting is exceedingly challenging: the agent provided only with examples of success states (e.g., an observation where an object has been placed in the correct location). Most prior methods that tackle similar tasks employ hand-designed reward functions or distance functions, full demonstrations, or carefully-constructed initial state distributions.

The results in Fig. 2 show that RCE significantly outperforms prior methods across all tasks. The transparent lines indicate one random seed, and the darker lines are the average across random seeds. RCE solves many tasks, such as bin picking and hammering, that *none* of the baselines make *any* progress towards solving. The most competitive baseline, SQIL, only makes progress on the easiest two tasks; even on those tasks, SQIL learns more slowly than RCE and achieves lower asymptotic return. To check that all baselines are implemented correctly, we confirm that all can solve a very simple reaching task described in the next section.

## 5.2 Example-Based Control from Images

Our second set of experiments studies whether RCE can learn image-based tasks and assesses the generalization capabilities of our method. We designed three *image-based* manipulation tasks. The `reach_random_position` task entails reaching a red puck, whose position is randomized in each episode. The `reach_random_size` task entails reaching a red object, but the actual shape of that object varies from one episode to the next. Since the agent cannot change the size of the object and the size is randomized from one episode to the next, it is impossible to reach any of the previously-observed success examples. To solve this task, the agent must learn a notion of success that is more general than reaching a fixed goal state. The third task, `sawyer_clear_image`, entails clearing an object off the table, and is mechanically more challenging than the reaching tasks.

Fig. 4 shows results from these image-based experiments, comparing RCE to the same baselines. We observe that RCE has learned to solve both reaching tasks, reaching for the object regardless of the location and size of the object. This task is mechanically easier than the state-based tasks in Fig. 2,

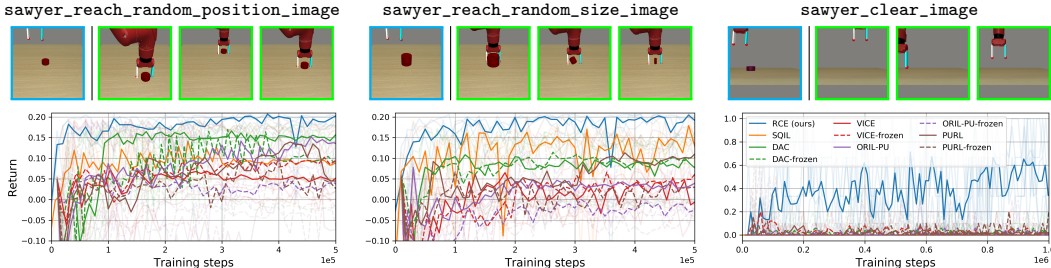

Figure 4: **Example-based control from images**: We evaluate RCE on three manipulation tasks using image-observations. *(Top)* We show examples of the initial state and success examples for each task. *(Bottom)* RCE (blue line) outperforms prior methods, especially on the more challenging clearing task. For the `random_size` task *(center)*, this entails reaching for new objects that have different sizes from any seen in the success examples.

and all the baselines make some progress on this task, but learn more slowly than our method. The good performance of RCE on the `reach_random_size` task illustrates that RCE can solve tasks in a new environment, where the object size is different from that seen in the success examples. We hypothesize that RCE learns faster than these baselines because it "cuts out the middleman," learning a value function directly from examples rather than indirectly via a separate reward function. To support this hypothesis, we note SQIL, which also avoids learning a reward function, learns faster than other baselines on these tasks. On the more challenging clearing task, only our method makes progress, suggesting that RCE is a more effective algorithm for learning these image-based control tasks. In summary, these results show that RCE outperforms prior methods at solving example-based control tasks from image observations, and highlights that RCE learns a policy that solves tasks in new environments that look different from any of the success examples.

## 5.3 Ablation Experiments

We ran seven additional experiments to study the importance of hyperparameters and design decisions. Appendix F provides full details and figures. These experiments highlight that RCE is not an imitation learning method: RCE fails when applied to full expert trajectories, which are typically harder to provide than success examples. Other ablation experiments underscore the importance of using n-step returns and validate the approximation made in Sec. 3.2.

## 6 Conclusion

In this paper, we proposed a data-driven approach to control, where examples of success states are used in place of a reward function. Our method estimates the probability of reaching a success example in the future and optimizes a policy to maximize this probability of success. Unlike prior imitation learning methods, our approach is end-to-end and does not require learning a reward function. Our method is therefore simpler, with fewer hyperparameters and fewer lines of code to debug. Our analysis rests on a new data-driven Bellman equation, where example success states replace the typical reward function term. We use this Bellman equation to prove convergence of our classifier and policy. We believe that formulating control problems in terms of data, rather than the reward-centric MDP, better captures the essence of many real-world control problems and suggests a new set of attractive learning algorithms.

**Limitations and future work.** One limitation with RCE is that, despite producing effective policies, the classifier's predictions are not well calibrated. This issue resembles the miscalibration in Q-functions observed in prior work [9, 20]. Second, both RCE and the baselines we compared against all struggled to learn more challenging image-based tasks, such as image-based versions of the tasks shown in Fig. 3. Techniques such as explicit representation learning [19, 23] may be important for scaling example-based control algorithms to high-dimensional tasks.

**Acknowledgements.** This work is supported by the Fannie and John Hertz Foundation, NSF (DGE1745016, IIS1763562), ONR (N000141812861), and US Army. We thank Ilya Kostrikov for providing the implementation of DAC. We thank Ksenia Konyushkova, Konrad Zolna, and Justin Fu for discussions about baselines and prior work. We thank Oscar Ramirez for help setting up the image-based experiments, Ryan Julian and anonymous reviewers for feedback on drafts of the paper, and Alex Irpan for help releasing code.

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
