# OpenReview forum: "Replacing Rewards with Examples: Example-Based Policy Search via Recursive Classification"
_NeurIPS.cc/2021/Conference — NeurIPS 2021 Oral_

### Official Review · Reviewer_LgSM · 2021-07-16

**Rating:** 8
**Confidence:** 4

**Summary:**

This paper proposes a novel RL algorithm that operates without an explicit reward function. Instead, the algorithm directly learns a value function from transitions and successful outcomes without ever learning an intermediate reward function. This approach also satisfies a new data-driven Bellman equation, where examples replace the reward function. The authors prove that this new Bellman equation converges to a Bayesian optimal classifier and optimal policy. By proving these results, this method retains the theoretical guarantees of the standard reward-based RL  algorithms, while simplying implementation and reducing the number of hyperparameters to be learned. The authors illustrate that their proposed approach performs significantly better than the reward-based RL algorithms on the 'Sawyer manipulation tasks'.

**Limitations And Societal Impact:**

The authors have adequately addressed the limitations of the their work and it poses no apparent negative societal impact.

**Main Review:**

**Pros**

* The paper is well written and easy to follow.
* Presenting a data-driven approach to RL is very useful, and this work opens avenues for future work in this area, mitigating the issue of carefully hand-crafting reward functions.
* By presenting strong theoretical convergence results with the new Bellman equation, the proposed algorithm successfully retains the properties of the existing RL algorithms while still bypassing the reward function. And this result makes the algorithm useful in a practical setting.
* The experimental evaluations are also comprehensive and illustrate the key ideas of the paper, on complex environments.

**Cons**

* Evaluation: This is not really a negative of the paper, but it would be interesting to see the performance of this approach in a setting where the definition of success of a task is non-binary.


**General Comments**
The paper is well written and the ideas presented are very clear. The experimental evaluations are comprehensive. This work opens avenues for future research in data-driven algorithms in RL.


*Originality*: Novel

*Clarity*:  Well written.

*Quality*: Good

*Significance*: High

UPDATE (Post Rebuttal): Thank You authors for your responses to the specific queries. After reading the other reviews and the authors' responses to them, I am updating my score.

**Time Spent Reviewing:**

3

---

> ### Author Response · Authors · 2021-08-06
> **Response to LgSM**
>
> Thank you for the detailed review.
>
> > a setting where the definition of success of a task is non-binary.
>
> We will investigate these settings. We welcome suggestions for specify settings to consider.

---

### Official Review · Reviewer_xycR · 2021-07-16

**Rating:** 9
**Confidence:** 4

**Summary:**

This is an interesting work in which the authors propose a new RL method to learn from examples instead of a pre-defined reward function. Existing inverse RL methods commonly introduce additional bias in estimating the reward function, which can be potentially addressed using the approach proposed in this paper.  In the experiments, the authors have demonstrated the effectiveness of this method in complex control tasks.

**Limitations And Societal Impact:**

This method can be potentially used in a wide range of real-world applications (e.g., automation control) where the reward function cannot be easily obtained.

**Main Review:**


Overall, I think this paper is very well presented and the work is solid. The authors have done extensive analysis and comparisons to show the effectiveness of the proposed method. I just have one minor comment. I agree that the estimation of reward function in IRL may degrade the performance, but it may also provide some interpretability (if the method is designed properly) on the expectation of the successful actions. Indeed, the proposed method has achieved better accuracy, but it may lose such interpretability. It would be great that the authors can comment on this.


**Time Spent Reviewing:**

2

---

> ### Author Response · Authors · 2021-08-06
> **Response to xycR**
>
> Thank you for the detailed review.
>
> > interpretability of the reward function
>
> Great point! Interestingly, we can post-hoc infer the reward function from the learned classifier. Specifically, given the learned classifier, we can rearrange the Bellman Equation (Eq. 11) to solve for the reward function term, $(1 - \gamma) p(e | s)$:
> $$\underbrace{(1 - \gamma) p(e = 1 \mid s)}_{\text{implicit reward function}} = \frac{C}{1 - C} - E\left[\frac{C'}{1 - C'} \right]$$,
> where $C'$ denotes the classifier evaluated at the state-action pair for time $t+1$.
>
> Thus, the reward function implicitly learned by RCE can be inspected in the same way that the explicit reward function learned by prior IRL methods can be inspected. We will include a visualization of the *implicit* reward function learned on the sawyer drawer task in the final paper.

---

### Official Review · Reviewer_6QWr · 2021-07-16

**Rating:** 7
**Confidence:** 4

**Summary:**

The paper proposes an approach for reinforcement learning in which the reward function is replaced by a recursive classification operation. More specifically, the method identifies if a success state will be reached. The set of success states is provided by an expert as a replacement for the reward function. The underlying rationale is that such information is much easier and more intuitive to provide that complex reward functions. The paper introduces the algorithm but also provides a rigorous formal discussion of its properties. In particular, proofs are provided that ensure that learned classifiers converge to the Bayes-optimal case and that the system converges to the optimal policy. Beyond that, the paper also discusses a variant of the method under stricter assumptions, called robust example-based control. Experiments in simulated manipulation tasks show that the method substantially outperforms previous methods.

**Limitations And Societal Impact:**

The authors convincingly discussed limitations to their approach, e.g., calibration of the classifier. No societal impact expected.

**Main Review:**

The main rationale behind this paper is not necessarily novel, other papers have already indicated that examples and outcomes can be used for improved RL. However, this paper consequently and rigorously follows this line of thought and discusses a number of exciting algorithmic and theoretical tricks to completely do away with the reward function. The derivation of the algorithm is carefully explained and detailed (even if not necessarily trivial or easy to understand). I also seems to me like the introduced framework can be reused by others to create a family of example-based policy search algorithms. For example, the Bayes classifier may be replace with a more powerful method down the road, as long as it satisfies similar theoretical conditions. The presented work is how very high quality and includes clear improvements along both the theoretical and practical dimension. The empirical results show a clear advantage over a number of SotA methods from very recent papers. The chosen tasks and domains are among the trickier 3D and physics-based benchmarks for RL. The paper is very well written and meticulous in it's presentation. The authors also do a great job in highlighting interesting insights they discovered along the way, i.e., the equivalence to Hellinger distance. Overall, the presentation is very clear.

Overall, I think that the introduced method is high significance to the NeurIPS community and that the authors did a great job in deriving the method and presenting it. In particular, the paper addresses different modifications of the method - generally a sign that the authors carefully analyzed their results from different vantage points and under different assumptions. The empirical results are pretty convincing, too. The only slightly confusing aspect is why the authors chose to formulate it in an off-policy fashion. Does this come with any drawbacks or is the decision purely to ensure that information from other policies can be re-used?

**Time Spent Reviewing:**

7

---

> ### Author Response · Authors · 2021-08-06
> **Response to 6QWr**
>
> Thank you for the detailed review.
>
> > Drawbacks of RCE being an off-policy method
>
> Our motivation for focusing on the off-policy setting was data efficiency. We plan to use RCE in robotic experiments in the future, so we wanted to use a sample-efficient algorithm (off-policy algorithms are generally more sample efficient than on-policy methods). The main drawback of using an off-policy RL algorithm is that it has a few more moving parts than on-policy methods.

---

### Official Review · Reviewer_9r4w · 2021-07-19

**Rating:** 8
**Confidence:** 3

**Summary:**

The paper describes a novel algorithm (RCE) to solve control tasks without an explicit reward function or expert trajectories. Rather, the method relies on a relatively small set of success (goal) states provided by a user. The paper is well written and offers intuitively clear algorithmic ideas which are novel, to my knowledge. The results on several manipulation tasks show strong performance, outperforming state-of-the-art IL baselines. Overall, the paper seems like a good contribution to the literature on learning control from expert data with the main difference that the proposed algorithm only requires a small sample of the distribution over goal states. My score primarily reflects my open questions and I'm open to increasing the score based on the authors response.

**Limitations And Societal Impact:**

The paper includes a brief discussion on its limitations and the authors state that they do not foresee plausible negative impacts of this work. That seems reasonable to me. Perhaps a future consideration might be how to ensure the data collection of success states is conducted well as end users will likely be involved.

**Main Review:**

- The key algorithmic idea is to learn a classifier whose prediction prob ratio is the expected prob that the current (s_t, a_t) will lead to a success state at some point in the future. The main challenge is sampling from successful trajectories, which are not available here. Practically, this means training the classifier to discriminate between state-action pairs on successful trajectories vs the rest, although the "negative" examples seem to potentially include successful (s, a) pairs. The loss function combines both types of data examples, which is used to update the classifier's parameters. The behavior policy is trained to maximize the classifier's output. Overall, the approach seems to be intuitively clear and the final algorithm is relatively straightforward. As far as I can tell, this is a novel approach to the problem of learning a policy from a sample distribution of goal states and the environment.

- While the main ideas of the paper are reasonably well described, I think the paper might become a bit easier to understand if a single running example was used throughout the problem setup. This would potentially have the benefits of making the core concepts of task, success state, etc. concrete up front, motivate the need to learn a generalized notion of success (vs simply memorizing the provided examples) as well as the need for assumptions on the user's state distribution in Section 3.1. reach_random_size seems like a candidate but there are probably better domains / tasks.

- Is learning a stopping criteria relevant here? It wasn't clear to me if and how well the agent learns to stop in success states. What's the action $a_t$ in the dataset $(s_t, a_t | e_t = 1)$? During evaluation (test), does the evaluation automatically end when the agent first reaches a success state? It seems to be a mixed bag in the videos (which are a very useful visualization of the learned behavior), where the agent seems to halt in some tasks but not in others (e.g., put the green object in the blue bin). Is the failure case (Pick up the ball) an instance of this issue or something else?

- The experiments are performed on a number of manipulation tasks. The user provides 200 success states for each of the manipulation tasks. The main novelty here is that these are individual states, not trajectories (which IL methods typically require). This is an impressive result and the supporting materials (videos, code, appendix) are quite detailed. However, the overall data requirements of RCE vs IL methods are not clear to me. RCE presumably uses much less expert data than IL but this could be quantified more clearly. A somewhat different set of questions involve analyzing the relatively poor performance of IL methods. Is the reason insufficient expert data? Something else? A more careful comparison and analysis of RCE vs IL methods would significantly improve the paper, in my opinion.

- Overall, I think the paper is well written, tackles important problems, and introduces new algorithmic ideas supported with analysis. The experiments are sufficiently detailed and demonstrate strong performance and outperform existing IL methods. While I think the empirical discussion and analysis could be improved, overall, I think this might be a good addition to the literature on learning control from (expert) data. My score reflects my open questions and I'm open to increasing the score based on the authors response.

UPDATE: I thank the authors for their detailed response to the reviewers. I think that most concerns about this paper have been well addressed at this point. I'm now more inclined to accept the paper.

**Time Spent Reviewing:**

8

---

> ### Author Response · Authors · 2021-08-06
> **Response to 9r4w**
>
> Thank you for the detailed and insightful review.
>
> > Is learning a stopping criteria relevant here?
>
> All our experiments used fixed horizon episodes; there is no special stopping criterion or HALT action. If the environment dynamics allow the agent to stop at a success state, then the agent will do so, because doing so maximizes the probability of future success. In settings where the agent is unable to stop at the success state, then it will aim to return to a success state as quickly as possible.
>
> > What's the action at in the dataset $(s_t, a_t \mid e_t=1)$?
>
> We do not assume that success examples are labeled with corresponding actions. Rather, we derive RCE by inferring the corresponding action using the behavior policy, $p(a_t \mid s_t)$ (see L533 - L534 in the Appendix for more discussion). Ablation experiments (L768 - L772, Fig 6d) show that sampling actions from the current policy, rather than the behavior policy (as theory prescribes) does not decrease performance. We therefore use the current policy for sampling actions in all other experiments.
>
> > does the evaluation automatically end when the agent first reaches a success state?
>
> No. Please see the videos on the project website that help visualize this.
>
> > Why doesn't the agent halt when putting the green object in the blue bin?
>
> All states where the object is in the correct bin are success states, so the policy incurs no penalty from moving the object around. Since we regularize the policy with a small entropy bonus (L204), the policy is incentivized to keep moving around.
>
> > Is the failure case (pick up the ball) an instance of [failing to halt] or something else?
>
> Our hypothesis is that RCE fails to learn this task because the exploration problem is too challenging. In future work, we plan to equip RCE with more sophisticated exploration strategies.
>
> > RCE presumably uses much less expert data than IL but this could be quantified more clearly
>
> Ablation experiments in Fig 6a indicate that RCE continues to perform well when using around 10 success examples. These tasks had a horizon of 151, so this is $10 / 151 = 0.07 \ll 1$ trajectories. To the best of our knowledge, even the best imitation learning methods require at least one full trajectory (15x more states than our method).
>
> > relatively poor performance of IL methods
>
> We offer two hypotheses to explain the poor performance of imitation learning methods. First, imitation learning methods were not designed for this problem setting: imitation learning methods assume access to expert trajectories (full sequences of states and actions) whereas our problem setting only provides success states (no transitions, no actions). Second, the tasks that we use are fairly complex, especially those tasks in Fig 3. To support this claim, we note that prior *reward-based* approaches to solve these tasks required careful reward engineering [1, 2]. The imitation learning baselines perform better on the image-based tasks in Fig 4, which are easier from a control perspective.
>
> [1] Fu, Justin, et al. "D4rl: Datasets for deep data-driven reinforcement learning." arXiv preprint arXiv:2004.07219 (2020).
> [2] Rajeswaran, Aravind, et al. "Learning complex dexterous manipulation with deep reinforcement learning and demonstrations." arXiv preprint arXiv:1709.10087 (2017).

---

### Decision · Program_Chairs · 2021-09-28

**Decision:**

Accept (Oral)

**Comment:**

This was a well-received paper by all the reviewers.  The discussion strengthened the acceptance score.

**Consistency Experiment:**

NeurIPS has a long history of experimentation. In 2014, NeurIPS ran an experiment in which 10% of submissions were reviewed by two independent committees to quantify the randomness in the review process. This year, we repeated a variant of this experiment to see how the quality of the review process has changed over time.  This paper was part of the experiment and was therefore assigned to two committees (consisting of reviewers, an Area Chair, and a Senior Area Chair) that reached independent decisions.  If both committees made the same recommendation, this recommendation was followed. If a single committee recommended acceptance, the paper was accepted (with the exception of a few cases in which the other committee identified what we considered a fatal flaw, e.g., an error in a key result).

This copy’s committee reached the following decision: **Accept (Oral)**

The other committee assigned to the paper recommended **Accept (Poster)**.  You can find the other set of reviews, along with any follow up discussion with the authors here:
https://openreview.net/forum?id=VXeoK3fJZhW